# MiR-506 Promotes Antitumor Immune Response in Pancreatic Cancer by Reprogramming Tumor-Associated Macrophages toward an M1 Phenotype

**DOI:** 10.3390/biomedicines11112874

**Published:** 2023-10-24

**Authors:** Tiantian Yang, Yitong Han, Junhang Chen, Xiaoyu Liang, Longhao Sun

**Affiliations:** Department of General Surgery, Tianjin Medical University General Hospital, Tianjin 300052, Chinaliang_xiaoyu@263.net (X.L.)

**Keywords:** miR-506, pancreatic cancer, reprogram, signal transducer and activator of transcription 3, tumor-associated macrophage

## Abstract

Pancreatic ductal adenocarcinoma (PDAC) is a highly malignant cancer with a poor prognosis, and effective treatments for PDAC are lacking. In this study, we hypothesized that miR-506 promotes antitumor immune response in PDAC by reprogramming tumor-associated macrophages toward an M1 phenotype to reverse its immunosuppressive tumor microenvironment (TME). First, the relationship between TME and the expression of miR-506 was assessed using clinical samples. Our results provided evidence that lower expression of miR-506 was associated with poor prognosis and immunosuppressive TME in PDAC patients. In addition, miR-506 inhibit the PDAC progression and reversed its immunosuppressive microenvironment in a macrophage-dependent manner. Next, we established a PDAC mouse model by orthotopic injection to further explore the role of miR-506 in vivo. Mechanistic investigations demonstrated that miR-506 could reprogram the polarization of M2-like macrophages toward an M1-like phenotype through targeting STAT3. Meanwhile, miR-506 could also sensitize PDAC to anti-PD-1 immunotherapy, because the tumor microenvironment remodeling effects of miR-506 could reprogram macrophage polarization and subsequently promote cytotoxic T lymphocyte (CTL) infiltration. These findings suggest a relationship between miR-506 and TME, especially M2-like macrophages, thus providing novel insights into mechanisms of tumor progression and potential immunotherapeutic targets for further clinical treatment.

## 1. Introduction

Pancreatic ductal adenocarcinoma (PDAC) is a highly aggressive malignancy. Its 5-year survival rate is <10% [1,2]. Surgical resection with chemotherapy remains the most effective therapeutic strategy [3,4], yet, considering that most patients present with late-stage cancer (tumor being either unresectable or metastatic), alternative treatment options are considered. While immunotherapy and targeted therapy have been well investigated in past decades, no therapy has been found to be fully effective [5]. Therefore, exploring novel therapeutic targets and biological markers for early diagnosis and treatment is urgently needed.

Unlike other types of cancers, the extensive desmoplastic stroma, which mainly refers to the TME rich in stroma cells and the deposition of extracellular matrix (ECM)*,* is a unique hallmark of PDAC [6,7]. The TME of PDAC consists of fibroblasts, neurons, endothelial cells, EMC proteins, and immune infiltration cells [8]. Studies have shown that immune cells have a vital role in the biology of TME in PDAC patients [9,10]. For example, macrophages, accounting for the highest proportion in the TME, including both resident and recruited circulating macrophages, have a critical role in the homeostasis of TME and tumor progression [11,12]. Furthermore, in the TME, macrophages can be transformed into an M1-like phenotype with anti-cancer immunity and M2-like phenotype with immunosuppressive function. These cells can repolarize and transform mutually due to the alteration of the TME [13,14]. M2-like macrophages, also known as tumor-associated macrophages (TAMs), possess potent immunosuppressive activity. Previous studies also demonstrated that M2 TAMs are strongly associated with tumor progression, metastasis, and drug resistance in PDAC patients [15,16], suggesting that TAMs could be potentially used as immunotherapeutic targets for clinical treatment with PDAC patients.

Signal transducer and activator of transcription 3 (STAT3) is a transcription factor that regulates a series of biological processes [17,18]. Hyperactivated STAT3 has been found in multiple cancers and has been associated with poor prognoses [19]. Activated STAT3 also affects immune cells in the TME by regulating the expression of several cytokines (interleukin 6 (IL-6) and IL-10) and vascular endothelial growth factor (VEGF) [20]. Moreover, these tumor-derived cytokines and factors can activate and enhance STAT3 signaling in the TME and tumor cells, which promotes tumor progression and impairs antitumor immunity [21,22,23].

MicroRNAs (miRNAs) are 20- to 25-nucleotide-long noncoding single-stranded RNA molecules. They can play an oncogenic or oncosuppressive role in various malignancies by regulating the expression of oncogenic or tumor-suppressive target genes. Notably, several miRNAs have been found to be upregulated or downregulated in various tumors, including PDAC, which can break down the mRNA transcript or inhibition of the translation of the mRNA to proteins implicated in cancer pathogenesis [24,25]. MiR-506 has been negatively associated with tumor T stage and lymph node metastasis, and its deregulation has been associated with pancreatic progression [26,27]. Moreover, another study proved that STAT3 is a direct target of miR-506 in PDAC [28].

In this study, we further explored the relationship between the expression of miR-506 and alteration of the TME in PDAC patients, revealing that miR-506 can promote antitumor immune response by reprogramming M2 macrophages to M1-like phenotype through targeting the STAT3 signaling pathway and enhancing the sensitivity of immunosuppressive therapy.

## 2. Materials and Methods

### 2.1. Clinical Specimens

Paired human PDAC and adjacent nontumor pancreatic tissue samples were collected from 126 patients with biopsy-proven PDAC at Tianjin Medical University General Hospital (TMUGH) between 2018 and 2021. Fresh surgical samples were processed into single-cell suspensions containing 2.5 U/mL hyaluronidase, 0.1 mg/mL DNase, 1 mg/mL collagenase. Samples were then subjected to flow cytometry; antibodies against human CD8, CD4, FOXP3, CD25, CD45, CD163, and CD68 (all BD Biosciences) were used as cell surface markers to identify human TAMs (CD68^+^/CD163^+^), M1 TAMs (CD68^+^/CD163^−^), Tregs (CD4^+^CD25^+^FOXP3^+^), M2 TAMs (CD68^+^/CD163^+^), cytotoxic T cells (CD8^+^/CD45^+^), and apoptotic T cells (annexin V^+^/CD8^+^) (all from BD Biosciences, San Jose, CA, USA). Enzyme-linked immunosorbent assay (ELISA) was used to analyze the IFN-γ in tumor tissue, according to the manufacturer’s procedure (R&D Systems, Minneapolis, MN, USA). Mirror-image tumor samples were collected and snap-frozen in liquid nitrogen and stored until RNA extraction. Clinical and biological information was registered for each patient, and follow-up data were recorded after informed consent was obtained. The institutional research ethics committee at TMUGH approved this study.

### 2.2. Macrophage Analysis

Density gradient centrifugation with Histopaque-1077 (Sigma-Aldrich, St. Louis, MI, USA) was used to obtain PBMCs isolated from healthy blood donors. CD14^+^ monocytes were then purified using CD14 microbeads (Miltenyi Biotech, Miltenyi Biotech) and cultured in RPMI 1640 medium (with 50 ng/mL GM-CSF (Peprotech, London, UK) and 20% heat-inactivated FBS (Sigma-Aldrich, St. Louis, MI, USA) for five days to differentiate into nonpolarized M0 macrophages.

### 2.3. Macrophage Polarization

On day 5, M0 macrophages were differentiated using M1 medium (50 ng/mL IFN-γ, 50 ng/mL GM-CSF, 100 ng/mL lipopolysaccharide, and 20 ng/mL TNF-α) or M2 medium (100 ng/mL M-CSF, 10 ng/mL IL-10, and 10 ng/mL TGF-β; Peprotech, London, UK), with miR-506 or miR-ctrl for 48 h. On day 7, polarized macrophages were analyzed using flow cytometry; antibodies against human MHC-II, CD86, CD163, CD204, CD80, and CD206 (all from BD Biosciences, San Jose, CA, USA) were used for cell surface markers.

### 2.4. Phagocytosis Assay

FITC-labeled *Escherichia coli* BioParticles (Thermo Fisher Scientific, Waltham, MA, USA) was mixed with polarized macrophages. Cells were then incubated with for 2 h in a humidified atmosphere containing 5% CO_2_/95% air at 37 °C. After removing the BioParticle loading suspension, trypan blue suspension was mixed with the solution. A microplate spectrophotometer was then used to analyze the relative phagocytosis.

### 2.5. Nitric Oxide (NO) Production

The Griess Reagent System (Promega, Madison, WI, USA) was used to analyze NO production. Briefly, a supernatant medium of different polarized macrophages was mixed with Griess reagent. The mixture was then incubated for ten min at 20 °C in the dark, after which the absorbance at 540 nm was measured using a microplate spectrophotometer; NO concentration was determined with sodium nitrite as a standard.

### 2.6. Arg1 Activity

Arg1 activity was analyzed as previously described [29]. Briefly, macrophages were lysed. Then, the supernatant was heated at 56 °C for 7 min to activate the Arg1. Then, the solution was mixed with 50 μmol of L-arginine for 2 h at 37 °C to allow hydrolysis of L-arginine by Arg1. The byproduct metabolite urea levels were determined by incubation with α-isonitrosopropiophenone substrate. The absorbance was measured at 540 nm, and the Arg1 urea production was calculated in samples from the urea standard curve.

### 2.7. VEGF, IL-10, and TGFβ Secretion

Polarized macrophages were cultured in RPMI 1640 + 20% heat-inactivated FBS for 24 h. The amount of VEGF, IL-10, and TGFβ was quantified using ELISA (R&D Systems) following the manufacturer’s procedure.

### 2.8. RNA Isolation and qRT-PCR

The MirVana Isolation Kit (Ambion, Austin, TX, USA, AM1560) was used to obtain the total RNA, including miRNA. Reverse transcription was performed using the TaqMan miRNA reverse-transcription kit (Applied Biosystems, Beverly Hills, CA, USA, 4366596) or SuperScript II reverse transcriptase (Invitrogen, Waltham, MA, USA, 18064014). TaqMan gene expression assay and TaqMan microRNA assay (Applied Biosciences, Beverly Hills, CA, USA, A25576 and 4331182, respectively) were used to detect and quantify mir-506 CD80, CD86, human MHCII, IL1B, TNFA, NOS2, CD163, CD204, ARG1, IL10, CD206, TGFB1, and STAT3. Relative expression was normalized to the endogenous control GAPDH or RNU6 using the 2^−ΔΔCt^ method. Experiments were carried out in triplicate.

### 2.9. Western Blotting

Cells were mixed with RIPA buffer (Thermo Fisher Scientific, Waltham, MA, USA, 89901). The whole-cell lysate was then separated on a 10% polyacrylamide gel. The membrane was then blocked in 5% nonfat milk in tris-buffered saline (TBS; BIO-RAD, Hercules, CA, USA, 1706435, pH7.6) with 0.1% Tween-20 (Sigma-Aldrich, St. Louis, MI, USA, P2287). Samples were then incubated with primary antibodies (anti-phospho-STAT3, Tyr705, Cell Signaling Technology, Danvers, MA, USA, 9145, 1:250; anti-STAT3, Cell Signaling Technology, Danvers, MA, USA, 12640, 1:500) at 4 °C overnight and then with secondary antibodies (Santa Cruz Biotechnology, Santa Cruz, CA, USA, sc-2077 or sc-2375 at a concentration of 1:10,000) at room temperature for 2 h. The proteins were visualized using SuperSignal West Pico chemiluminescent substrate or Femto Maximum Sensitivity Substrate (Thermo Fisher Scientific, Waltham, MA, USA, 34080 or 34095).

### 2.10. Reagents and Antibodies

SiRNAs for STAT3 (SASI_Hs01_00121206 and 00121207) were from Sigma. The miRNA mimics of miR-506 and control miRNA (miR-ctrl) were bought from Dharmacon (c-300846-05 and CN-001000-01). Briefly, cells were seeded in 6-well plates (2 × 10^5^/well) overnight, after which they were transfected with miR-ctrl, miR-506 mimic, or siRNA for STAT3 using lipofectamine RNAiMAX (Invitrogen, Carlsbad, CA, USA, 13778150). Total RNA and protein were collected 48 h after transfection.

### 2.11. Orthotopic PDAC Mouse Model

Female-specific pathogen-free (SPF) C57BL/6 mice (age, 4 weeks; weight, 20–22 g) were housed in a specific pathogen-free environment with relative humidity of 50 ± 1%, temperature of 22 ± 1 °C, and light/dark cycle of 12/12 h. All animal studies were done in compliance with the regulations and guidelines of Tianjin Medical University institutional animal care and conducted according to the AAALAC and the IACUC guidelines.

The mouse PDAC cell line Panc02 (2 × 10^6^), purchased from ATCC (Rockville, MD, USA), was injected into the pancreas of C57BL/6 mice. Animals were then randomly divided into two groups (n = 10 per group): miR-ctrl group and miR-506 group. MiRNAs were injected intraperitoneally for 4 weeks. Next, mice were euthanized, and the xenograft tumors were harvested and weighed. The tumors were dissociated into single-cell suspensions and analyzed using flow cytometry analysis (antibodies against mouse M2 TAMs (F4/80^+^/CD206^+^), M1 TAMs (F4/80^+^/CD206^−^), Tregs (CD4^+^CD25^+^FOXP3^+^), cytotoxic T cells (CD8^+^/CD45^+^), and apoptotic T cells (annexin V^+^/CD8^+^) (BD Biosciences, San Jose, CA, USA) were used. IFN-γ in mouse tumor tissue was measured by ELISA.

### 2.12. Statistical Analysis

SPSS 17.0 software (SPSS Inc., Chicago, IL, USA) was used for all statistical analyses. Data are expressed as the mean ± SD of at least 3 separate experiments performed in triplicate. Differences between groups were analyzed using Student’s *t*-test. A *p* value < 0.05 was considered to be statistically significant.

## 3. Results

### 3.1. Low Endogenous miR-506 Levels Are Associated with Poor Outcomes in PDAC Patients

To identify the role of miR-506 in the progression of PDAC, we evaluated the correlation between clinical stages and miR-506 expression level in a cohort of 126 patients. Relative miR-506 expression levels were higher in adjacent normal pancreas tissues than in PDAC tissues (*** *p* < 0.001; Figure 1A). Clinicopathological analyses further showed an inverse correlation between the level of miR-506 and T (primary tumor size) and TNM stage. Additionally, patients with early-stage tumors had higher miR-506 expression compared to those with advanced tumor (expression was lower in the T3–T4 group and III–IV group than in the T1–T2 group and I–II group) (*** *p* < 0.001, Figure 1B,C). Furthermore, the patients with low miR-506 showed more advanced T and TNM stages than those with high miR-506 levels (*** *p* < 0.001, Figure 1D,E). Finally, Kaplan–Meier analysis indicated significantly shorter OS in patients with a low level of miR-506 within the tumor compared to those with high miR-506 expression (*p* < 0.001; Figure 1F). All of these results suggest that low miR-506 expression may be used as a biomarker for poor prognosis in PDAC patients.

### 3.2. Low Endogenous miR-506 Levels Are Associated with Immunosuppressive Microenvironment in Human PDAC

Twenty fresh tissue samples of human PDAC (10 in the miR-506 low group and 10 in the high miR-506 group) were assessed using flow cytometry to investigate the relationship between miR-506 level and infiltration of immune cells in the human PDAC microenvironment. Significantly higher percentages of CD4^+^CD25^+^FOXP3^+^ Tregs (*** *p* < 0.001; Figure 2B), higher ratios of M2 (CD68^+^CD163^+^)/M1 (CD68^+^CD163^−^) TAMs (*** *p* < 0.001; Figure 2A), fewer CD8^+^CD45^+^ cytotoxic T cells (*** *p* < 0.001; Figure 2C), a greater percentage of apoptotic cytotoxic T cells (*** *p* < 0.001; Figure 2D), and low production of IFN-γ by cytotoxic T cells (*** *p* < 0.001; Figure 2E) in the PDAC microenvironment were found in patients with low miR-506 levels. These data indicate that low miR-506 in PDAC correlates with an immunosuppressive TME.

### 3.3. MiR-506 Inhibits the PDAC Progression and Reverses Its Immunosuppressive Microenvironment in a Macrophage-Dependent Manner

An orthotopic PDAC mouse model was first constructed by injecting murine Panc02 cells into C57BL/6 mice. Animals were consequently treated with miR-ctrl or miR-506 for 4 weeks, after which the effect of miR-506 on the immune microenvironment and tumor progression were analyzed. Larger and heavier tumors and a higher percentage of Ki-67-positive PDAC cells were seen in the miR-ctrl group vs. the miR-506 group (*** *p* < 0.001; NS, not statistically significant; Figure 3A–D). In addition, miR-506 treatment significantly prolonged the OS (*p* < 0.001; Figure 3E).

Furthermore, flow cytometry analysis was performed on cell suspensions from the fresh mouse tumor tissues to investigate the relationship between immune cell infiltration in the tumor microenvironment and miR-506. M2 (F4/80^+^/CD206^+^)/M1 (F4/80^+^/CD206^−^) The TAM ratio was significantly reduced after the addition of MiR-506 (*** *p* < 0.001; NS, not statistically significant; Figure 3F). To further determine the role of macrophages in miR-506-mediated antitumor immunity and tumor suppression, we treated orthotopic tumor-bearing mice with clodronate liposomes (Clo) or PBS and miR-ctrl or miR-506 (n = 10 per group). MiR-506 significantly increased the infiltration of CD8^+^CD45^+^ cytotoxic T cells (*** *p* < 0.001; NS, not statistically significant; Figure 3H), decreased the infiltration of CD4^+^CD25^+^FOXP3^+^ Tregs (*** *p* < 0.001; NS, not statistically significant; Figure 3G), reduced the percentage of apoptotic cytotoxic T cells (*** *p* < 0.001; NS, not statistically significant; Figure 3I) and increased the production of IFN-γ by cytotoxic T cells (*** *p* < 0.001; NS, not statistically significant; Figure 3J) in the mouse PDAC microenvironment. However, we found that macrophage depletion by Clo significantly overturned the reversion of miR-506 on the immunosuppression of the TME. Macrophage depletion effectively annulled the differences in the percentage of Tregs (CD4^+^CD25^+^FOXP3^+^), cytotoxic T cells (CD8^+^CD45^+^), apoptotic cytotoxic T cells, and the production of IFN-γ by cytotoxic T cells between the miR-ctrl group and miR-506 group (*** *p* < 0.001; NS, not statistically significant; Figure 3G–J). To sum up, these results demonstrate that miR-506 inhibits PDAC progression and reverses its immunosuppressive microenvironment in a macrophage-dependent manner.

### 3.4. MiR-506 Reprograms the Polarization of M2 Macrophages

M1 or M2-polarized macrophages were then transfected with miR-506 mimic or miR-ctrl to further assess the impact of miR-506 on TAM polarization. The expression of alternative activated M2 surface markers (CD163, CD206, and CD204) was significantly decreased in M2-polarized macrophages transfected with miR-506 mimic compared with those transfected with miR-ctrl, while most of the classically activated M1 surface markers (MHC-II, CD86, and CD80) were upregulated. The M2-polarized macrophages were reprogrammed towards M1 phenotypes by miR-506. On the other hand, miR-506 could not affect the polarization of M1 macrophages (Figure 4A).

In addition, when M2-polarized macrophages were transfected with miR-506 mimic and compared with miR-ctrl, the expression of M2 genes (CD163, CD204, CD206, TGFB1, ARG1, and IL10) was downregulated, while the expression of M1 genes ((HLA-DRA, NOS2, IL1B, CD80, CD86, and TNFA) was increased (*** *p* < 0.001; NS, not statistically significant; Figure 4B). Furthermore, functional studies showed that when M2-polarized macrophages were transfected with miR-506 mimic, they displayed a shift toward a tumor-inhibition phenotype. MiR-506 transfection promoted nitric oxide (NO) production and phagocytosis of immunosuppressive M2-polarized macrophages while suppressing the IL-10, VEGF, TGF-β expression, and Arg1 activity. MiR-506 reversed the functions of M2-polarized macrophages, but not M1-polarized macrophages, toward cancer-suppressive activity (*** *p* < 0.001; NS, not statistically significant; Figure 4C–H).

### 3.5. STAT3 Is a Direct Target of miR-506 Involved in Macrophage Polarization

The expression level of miR-506 between different macrophage subtypes was analyzed to establish why miR-506 only affects the polarization of M2-polarized macrophages. qRT-PCR analyses revealed that miR-506 levels were significantly upregulated in M1 but not M2 macrophages. This differential expression suggested that miR-506 could be a negative regulator of alternative polarization of macrophages. To examine whether miR-506 contributes to the plasticity of macrophage polarization, we used different conditioned media trying to reprogram the subpopulation. When culturing macrophages with a conditioned medium, cells were repolarized from the M1 phenotype to the M2 phenotype. The M1-to-M2 conversion decreased the M1 macrophage markers and increased the expression of M2 macrophage markers, which resulted in an increased expression of miR-506. Conversely, M1-conditioned medium drove M2 macrophage repolarization to M1; the M2-to-M1 conversion increased the M1 markers and decreased the expression of M2 markers, which also decreased miR-506 expression (*** *p* < 0.001; NS, not statistically significant; Figure 5A,B). These results indicate that miR-506 is dynamically changing according to macrophage phenotype and may drive the repolarization of M2 macrophages with low miR-506.

Next, the 3′-untranslated region (UTR) of STAT3 mRNA was predicted to contain a potential binding site of miR-506; this site is highly conserved among different species (Figure 5C). Furthermore, STAT3 is involved in multiple aspects of mechanisms governing tumor escape from immune surveillance and is a direct target of miR-506 in PDAC cells. qRT-PCR revealed that the expression of STAT3 was different in a subtype of macrophages and inversely correlated with miR-506. Additionally, STAT3 levels were significantly more downregulated in M1 than in M2 macrophages (*** *p* < 0.001; NS, not statistically significant; Figure 5D), which validated why miR-506 only reverses the polarization of M2-polarized macrophages

Transfection with miR-506 mimics significantly reduced STAT3 levels in M2 macrophages. In contrast, no significant change in the expression levels of STAT3 was observed for M1 macrophage (*** *p* < 0.001; NS, not statistically significant; Figure 5E). To examine the regulation of STAT3 by miR-506, a luciferase reporter assay was performed. miR-506 significantly suppressed the luciferase reporter activity of the wild-type but not the mutant STAT3 3′-UTR, whereas anti-miR-506 treatment increased the luciferase activity of STAT3, indicating that STAT3 is a direct target of miR-506 (*** *p* < 0.001; NS, not statistically significant; Figure 5F).

### 3.6. MiR-506 Reprograms the Polarization of M2 Macrophage through the STAT3 Signaling Pathway

STAT3 is critical in tumor immune evasion and is persistently activated in most malignancies. STAT3 overactivation in tumor-infiltrating immune cells impairs antitumor immunity by compromising native immune responses via multiple mechanisms, including polarized macrophages toward the M2 phenotype. Loss-of-function studies were carried out to confirm that repolarization of M2 macrophages triggered by miR-506 was mediated by repression of STAT3.

First, we treated macrophages with a selective STAT3 inhibitor (Stattic). Following the decline of phosphorylated STAT3 (p-STAT3; Y705) expression, iNOS increased and Arg1 decreased in M2 macrophages, while the expression of p-STAT3, iNOS, and Arg1 did not significantly change in M1 macrophages. In addition, qRT-PCR analyses revealed decreased expression of alternative activated M2 phenotype genes and upregulation of most M1 phenotype genes were upregulated in M2 macrophages after Stattic treatment. In contrast, there was no significant change in the expression of phenotype genes in M1 macrophages (*** *p* < 0.001; NS, not statistically significant; Figure 6A top).

Silencing of STAT3 in M2 macrophages by small nterfering RNA (siRNA) (si-STAT3-2 and si-STAT3-1) significantly upregulated the expression of iNOS and downregulated the expression of Arg1, STAT3, and p-STAT3. Additionally, siRNA knockdown of STAT3 suppressed M2 genes and promoted most of the M1 genes compared to miR-ctrl, which reprogrammed M2 macrophages polarized toward the tumor-suppressive M1 phenotype (*** *p* < 0.001; NS, not statistically significant; Figure 6A bottom).

Next, we randomly treated an orthotopic PDAC mouse with vehicle or Stattic at 3.75 mg/kg/day to investigate whether STAT3 suppression modulates tumor progression and the immune microenvironment in vivo. After treatment, smaller and lighter tumors and a lower percentage of Ki-67-positive PDAC cells were seen in the Stattic group vs. the control group (** *p* < 0.01, *** *p* < 0.001; Figure 6B–D). Stattic treatment also prolonged OS in an independent cohort of mice (n = 20 per group; *p* < 0.001; Figure 6E).

Next, we performed a flow cytometry analysis. A significantly decreased percentage of apoptotic cytotoxic T cells, lower M2 (F4/80^+^/CD206^+^)/M1 (F4/80^+^/CD206^−^) TAM ratio decreased infiltration of CD4^+^CD25^+^FOXP3^+^ Tregs, increased infiltration of CD8^+^CD45^+^ cytotoxic T cells, and increased production of IFN-γ by cytotoxic T cells in the PDAC microenvironment were seen after Stattic treatment (** *p* < 0.01, *** *p* < 0.001; Figure 6F–J).

### 3.7. MiR-506-Induced Repolarization of Macrophages Sensitizes Them to Anti-PD-1 Immunotherapy

Anti-PD-1 immune checkpoint blockade therapy can suppress PD-L1-induced T cell inhibition to strengthen the antitumor immune response. However, anti-PD-1 immunotherapy is unlikely to be effective in PDAC because human pancreatic cancer exhibits very low CTL infiltration in the tumor microenvironment. Our results indicated that the tumor microenvironment remodeling effects of miR-506 reprogram macrophage polarization and subsequently promote CTL infiltration. Next, we evaluated whether miR-506 sensitizes pancreatic cancer to anti-PD-1 immunotherapy. Orthotopic PDAC mouse were treated with miR-ctrl iso, miR-ctrl PD1, miR-506 iso, and miR-506 PD1 (n = 10 per group). The analysis of the tumors indicated that combined miR-506 and anti-PD-1 immunotherapy have significantly greater efficacy than either miR-506 or anti-PD-1 Ab alone. Furthermore, smaller and lighter tumors, a lower percentage of Ki-67-positive PDAC cells, and more production of IFN-γ by CTLs in the tumor microenvironment were seen in the miR-506 PD1 group vs. other groups (n = 10 per group; ** *p* < 0.01, *** *p* < 0.001; Figure 7A–D). As expected, miR-506 + anti-PD-1 Ab treatment also significantly prolonged OS in an independent cohort of mice (n = 15 per group; ** *p* < 0.01, *** *p* < 0.001; Figure 7E).

## 4. Discussion

In the present study, we combined clinical data and basic experiments to explore the relationship between miR-506 and macrophages of TME in PDAC patients. First, we discovered that lower expression of miR-506 was associated with poor prognosis and immunosuppressive TME in patients with PDAC. Then, to further explore the detailed mechanism, we established the PDAC mouse model and found that miR-506 could reprogram the polarization of M2-like macrophages by targeting the STAT3 signaling pathway. MiR-506 could also sensitize PDAC to anti-PD-1 immunotherapy. Unlike previous studies, this study first investigated the miR-506 and TME, especially M2-like macrophages, thus providing novel insights into mechanisms of tumor progression and potential immunotherapeutic targets for further clinical treatment.

PDAC is a highly aggressive lethal malignancy due to limited treatment response and a lack of early diagnosis. Although adjuvant chemotherapy or neoadjuvant therapy has a significant survival benefit in patients with resected PDAC, most patients fail to receive therapy due to poor patient performance status, postoperative complications, or early disease progression [30]. Therefore, researchers are gradually turning to immune-targeted therapy to treat PDAC. Studies have found that multiagent cytotoxic regimens significantly affect advanced PDAC [31,32]. This approach requires activating the host immune system through cytotoxic T cells, which recognize tumor antigens [33]. However, in most PDAC patients, the TME is characterized by the absence of CD8^+^ T cells and abundant macrophages (especially TAMs) and regulatory T cells. These unique characteristics enhance the immunosuppression and drug resistance of PDAC, pointing to the need for effective and novel therapeutic targets for PDAC patients.

Macrophages originate from bone marrow hematopoietic stem cells (HSCs), which then differentiate into monocytes in the blood. During injury, monocytes are recruited and differentiated into macrophages with great plasticity [34]. Macrophages can differentiate into proinflammatory M1-like phenotypes with antitumor immune responses through classically activated pathways and anti-inflammatory M2-like phenotypes with immunosuppressive responses through alternatively activated pathways [35,36,37]. Meanwhile, fully polarized subgroups can repolarize and transform due to different TMEs [38]. Studies have shown that the TME of PDAC is abundant in M2 macrophages in patients with advanced PDAC, promoting tumor progression and metastasis [39,40]. Related clinical trials also proved that higher M macrophage infiltration is strongly correlated with a poorer prognosis and shorter OS. The density of M2 macrophage infiltration could be an independent prognostic factor for PDAC patients [41].

MiR-506 has an important role in tumorigenesis [42,43]. Yang et al. found that the lower expression of miR-506 in ovarian cancer patients is associated with poor prognosis [44]. Subsequent studies confirmed the tumor-suppressive function of miR-506 in other types of cancer [44,45,46,47,48]. As a multiagent regulatory gene, miR-506 can modulate cell differentiation through STAT3, flotillin 1, cell adhesion and migration through integrin beta 1 and 3, cell proliferation through cyclin-dependent kinase 4 and 6, and so on [49]. Among these targeted factors, STAT3 is critical for cell differentiation, proliferation, and angiogenesis [17].

STAT3 has been demonstrated to be a key mediator in contributing to the local immunosuppressive TME because of its activation within various immunosuppressive cells, including TAMs. Our previous study established that activated STAT3 can augment the expression and secretion of its downstream mediator IL-10, which promoted TAM polarization toward an M2 phenotype. Inhibiting STAT3 activation in tumor cells triggers macrophage production of NO and leads to macrophage-mediated, nitrite-dependent cytostatic activity against tumor cells. Our previous research confirmed that miR-506 can induce cell death through the STAT3 signaling pathway in PDAC patients. In this study, we further found that miR-506 can also repolarize M2-like macrophages, acting as a tumor-suppressive regulatory molecule by targeting the STAT3 pathway.

## 5. Conclusions

To sum up, our data suggest that miR-506 can reprogram the polarization of M2-like macrophages toward an M1-like phenotype through targeting STAT3. MiR-506 can also sensitize PDAC to anti-PD-1 immunotherapy and overcome immunotherapy resistance in PDAC because of the tumor microenvironment remodeling effects of miR-506 reprogramming macrophage polarization and subsequently promoting CTL infiltration. These findings suggest a strong association between miR-506 and tumor stage, prognosis, and drug sensitivity of PDAC, thus providing novel insights into mechanisms of tumor progression and potential immunotherapeutic targets for further clinical treatment. These data provide a rationale for clinical trials to evaluate the combination therapy of miR-506 and immune checkpoint therapy, particularly in immunotherapy-resistant tumors, such as PDAC, which lack infiltration of CTLs. However, this study on the method of administration, treatment process, dosage, toxicity and side effects of combination therapy needs further verification.

## Figures and Tables

**Figure 1 biomedicines-11-02874-f001:**
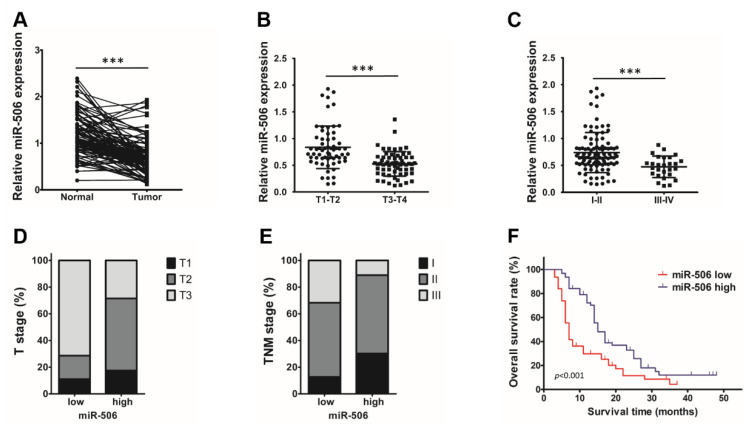
MiR-506 downregulation is correlated with disease progression in human PDAC. (**A**) Comparison of miR-506 expression in matched pairs of PDAC tissues and corresponding nontumor tissues via qRT-PCR. RNU6 was used as an internal control (n = 126; *** *p* < 0.001 by Student’s *t*-test). (**B**) Comparison of miR-506 expression in patients with stage T1–T2 and stage T3–T4 PDAC tissues (n = 126; *** *p* < 0.001 by Student’s *t*-test). (**C**) Comparison of miR-506 expression in patients with stage I–II and stage III–IV PDAC tissues (n = 126; *** *p* < 0.001 by Student’s *t*-test). (**D**) The distribution of the T stage between miR-506 high- and low-expression groups (n = 126). (**E**) The distribution of TNM stage between miR-506 high- and low-expression groups (n = 126). (**F**) Kaplan–Meier curves comparing the overall survival rates of PDAC patients whose tumors expressed a low or high level of miR-506. The median value was used as the cutoff point for the definition of low and high miR-506-expression groups (n = 126; *p* < 0.001 by log-rank test). Data represent the mean ± SD of at least 3 independent experiments.

**Figure 2 biomedicines-11-02874-f002:**
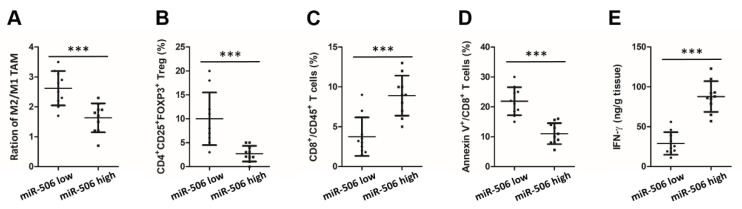
MiR-506 downregulation is correlated with immunosuppressive microenvironment in human PDAC. (**A**) Flow cytometry analysis of M2/M1 TAMs ratio in PDAC tissues from fresh surgical samples with low and high expression of miR-506 (CD68^+^CD163^−^ cells as M1 TAMs and CD68^+^CD163^+^ cells as M2 TAMs). (**B**–**D**) Flow cytometry analysis of CD4^+^CD25^+^FOXP3^+^ Tregs (**B**), CD8^+^CD45^+^ T cells (**C**), and apoptotic CD8^+^ T cells (**D**) in PDAC tissues from the same samples. (**E**) Production of IFN-γ by cytotoxic T cells in the PDAC microenvironment from the same patient samples analyzed by ELISA (n = 10 per group; *** *p* < 0.001 by Student’s *t*-test). Data represent the mean ± SD of at least 3 independent experiments.

**Figure 3 biomedicines-11-02874-f003:**
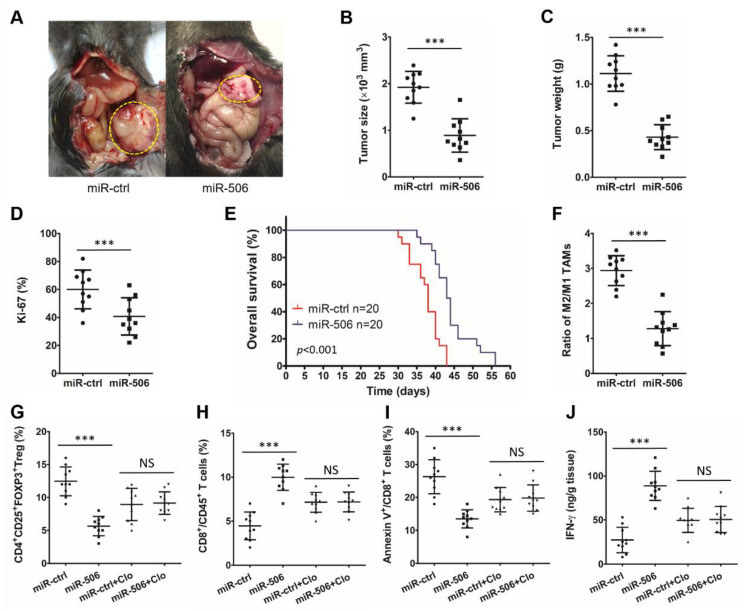
MiR-506 inhibits the tumor progression and reprograms its immunosuppressive microenvironment in a macrophage-dependent manner. (**A**) Panc02 cells were injected into C57BL/6 mice to establish an orthotropic PDAC model and then treated with miR-ctrl or miR-506. Twenty-eight days following cell implantation, mice were euthanized and tumors were harvested. (**B**–**D**) The tumor size (**B**), tumor weight (**C**), and percentage of mitotic Ki-67 tumor cells (**D**) were measured (n = 10 per group; *** *p* < 0.001 by Student’s *t*-test). (**E**) Kaplan–Meier curves comparing overall survival rates between miR-ctrl and miR-506 groups (n = 20 per group; *** *p* < 0.001 by log-rank test). (**F**) M2/M1 TAM ratio in the PDAC microenvironment from both groups of mice (F4/80^+^/CD206^−^ cells as M1 TAMs and F4/80^+^/CD206^+^ cells as M2 TAMs) analyzed by flow cytometry (n = 10 per group; *** *p* < 0.001 by Student’s *t*-test). (**G**–**I**) CD4^+^CD25^+^FOXP3^+^ Tregs (**G**), CD8^+^CD45^+^ T cells (**H**), and apoptotic CD8^+^ T cells (**I**) analyzed by flow cytometry. (**J**) Production of IFN-γ by cytotoxic T cells in the PDAC microenvironment from both groups of C57BL/6 mice analyzed by ELISA (n = 10 per group; *** *p* < 0.001; NS, not statistically significant by Student’s *t*-test). Data represent the mean ± SD of at least 3 independent experiments.

**Figure 4 biomedicines-11-02874-f004:**
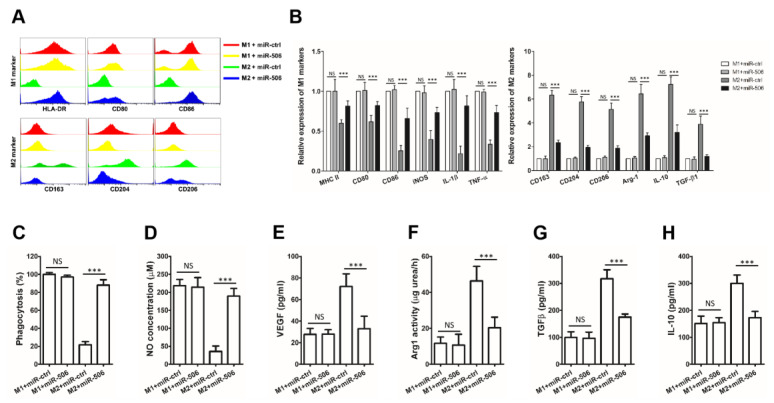
MiR-506 reprograms the polarization of M2 macrophages. (**A**) Human macrophage differentiation and polarization. Human monocytes were differentiated into nonpolarized M0 macrophages and then polarized into M1 or M2 macrophages using a conditioned medium. Polarized macrophages were treated with miR-ctrl or miR-506, and the expression levels of M1 and M2 macrophage surface markers were detected by flow cytometry. (**B**) The expression levels of M1 and M2 macrophage phenotypic genes for polarized macrophages were detected by qRT-PCR and normalized to the endogenous control GAPDH (*** *p* < 0.001; NS, not statistically significant by Student’s *t*-test). (**C**–**H**) Macrophage phagocytosis, nitric oxide (NO) production, IL-10, VEGF, TGF-β expression, and Arg1 activity measurements were established to characterize the functions of the miR-506 experimentally polarized macrophages (*** *p* < 0.001; NS, not statistically significant by Student’s *t*-test). Data represent the mean ± SD of at least 3 independent experiments.

**Figure 5 biomedicines-11-02874-f005:**
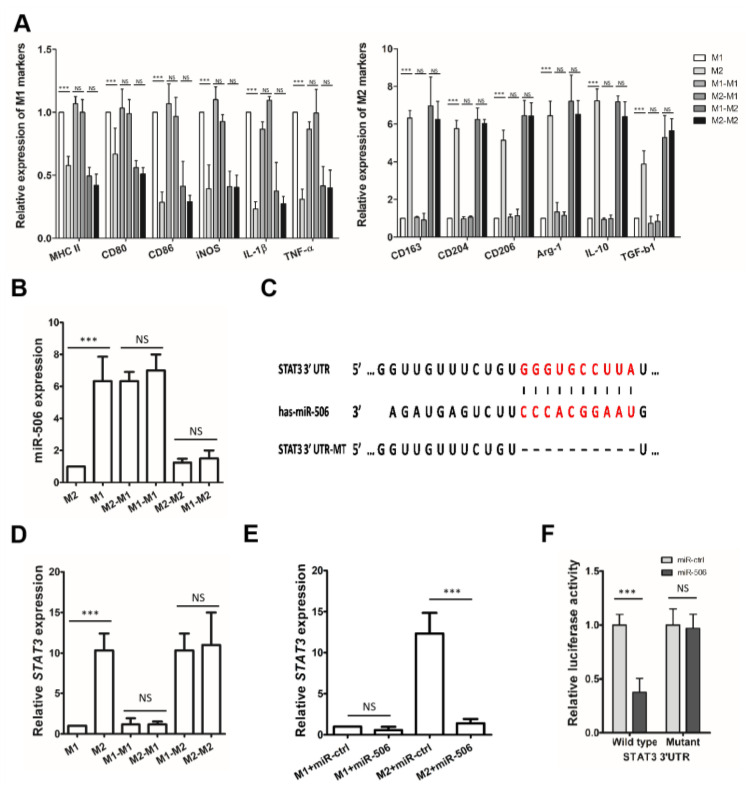
STAT3 is a direct target of miR-506 involved in macrophage polarization. (**A**) The expression levels of M1 and M2 macrophage phenotypic genes for repolarized macrophages detected by qRT-PCR and normalized to the endogenous control GAPDH (*** *p* < 0.001; NS, not statistically significant by Student’s *t*-test). (**B**) The expression levels of miR-506 for repolarized macrophages detected by qRT-PCR (*** *p* < 0.001; NS, not statistically significant by Student’s *t*-test). (**C**) An miR-506 binding site predicted in the 3′-UTR of STAT3 by TargetScan. (**D**) The expression levels of STAT3 for repolarized macrophages detected by qRT-PCR and normalized to the endogenous control GAPDH (*** *p* < 0.001; NS, not statistically significant by Student’s *t*-test). (**E**) Polarized macrophages were transfected with miR-ctrl or miR-506. The expression levels of STAT3 for repolarized macrophages detected by qRT-PCR and normalized to the endogenous control GAPDH (*** *p* < 0.001; NS, not statistically significant by Student’s *t*-test). (**F**) A luciferase reporter assay showed that miR-506 directly targeted the STAT3 3′-UTR (*** *p* < 0.001; NS, not statistically significant by Student’s *t*-test). Data represent the mean ± SD of at least 3 independent experiments.

**Figure 6 biomedicines-11-02874-f006:**
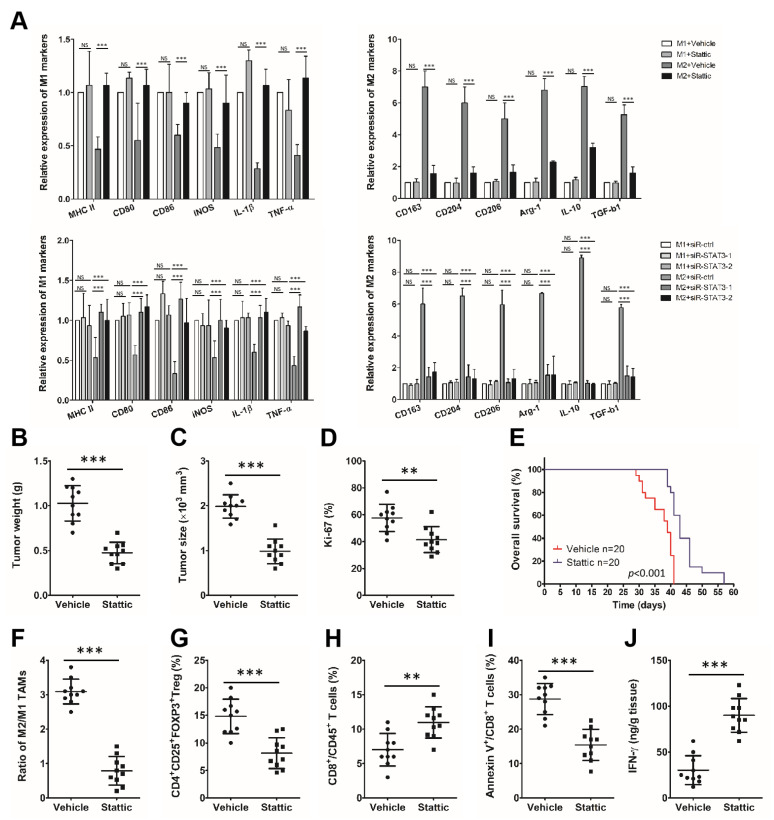
MiR-506 reprograms the polarization of M2 macrophages through the STAT3 signaling pathway. (**A**) The expression levels of macrophage phenotypic genes for macrophages repolarized by inhibiting or knocking down STAT3 detected by qRT-PCR and normalized to the endogenous control GAPDH (*** *p* < 0.001; NS, not statistically significant by Student’s *t*-test). (**B**) Panc02 cells were injected into C57BL/6 mice to establish an orthotropic PDAC model and then treated with Stattic or vehicle. Twenty-eight days following cell implantation, mice were euthanized and tumors were harvested. (**B**–**D**) The tumor weight (**B**), tumor size (**C**), and percentage of mitotic Ki-67 tumor cells (**D**) were measured (n = 10 per group; ** *p* < 0.01; *** *p* < 0.001 by Student’s *t*-test). (**E**) Kaplan–Meier curves comparing overall survival rates between Stattic and vehicle groups (n = 20 per group; *p* < 0.001 by log-rank test). (**F**) M2/M1 TAM ratio in the PDAC microenvironment from both groups of mice (F4/80^+^/CD206^−^ cells as M1 TAMs and F4/80^+^/CD206^+^ cells as M2 TAMs) analyzed by flow cytometry. (**G**–**I**) CD4^+^CD25^+^FOXP3^+^ Tregs (**G**), CD8^+^CD45^+^ T cells (**H**) and apoptotic CD8^+^ T cells (**I**) analyzed by flow cytometry. (**J**) Production of IFN-γ by cytotoxic T cells in the PDAC microenvironment from both groups of C57BL/6 mice analyzed by ELISA (n = 10 per group; ** *p* < 0.01; *** *p* < 0.001 by Student’s *t*-test). Data represent the mean ± SD of at least 3 independent experiments.

**Figure 7 biomedicines-11-02874-f007:**
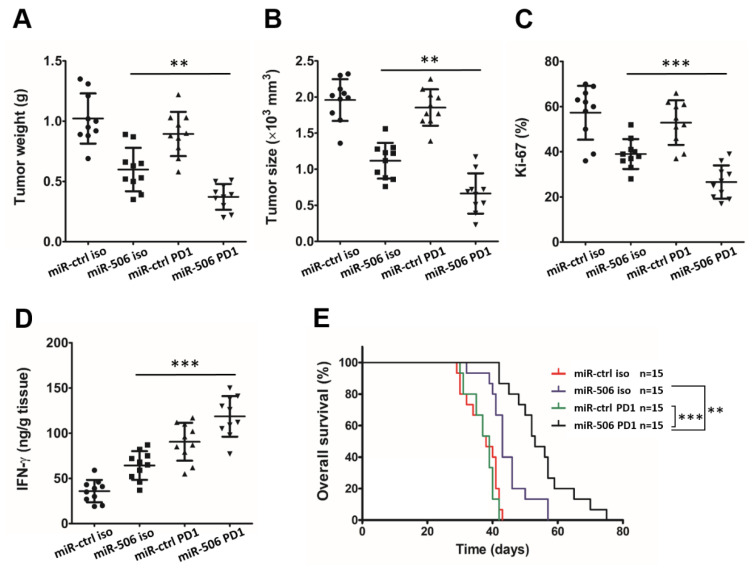
MiR-506 induced repolarization of M2 macrophage sensitizes pancreatic cancer to anti-PD1 therapy. (**A**) Panc02 cells were injected in C57BL/6 mice to establish an orthotropic PDAC model and then treated with miR-ctrl or miR-506 with or without anti-PD-1 therapy. Twenty-eight days following cell implantation, mice were euthanized, and tumors were harvested. (**A**–**C**) The tumor weight (**A**), tumor size (**B**), and percentage of mitotic Ki-67 tumor cells (**C**) were measured. (**D**) Production of IFN-γ by cytotoxic T cells in the PDAC microenvironment from both mouse groups analyzed by ELISA (n = 10 per group; ** *p* < 0.01; *** *p* < 0.001 by Student’s *t*-test). (**E**) Kaplan–Meier curves comparing overall survival rates between groups (n = 15 per group; ** *p* < 0.01; *** *p* < 0.001 by log-rank test). Data represent the mean ± SD of at least 3 independent experiments.

## Data Availability

Not applicable.

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
