# Peer review of "MiR-506 Promotes Antitumor Immune Response in Pancreatic Cancer by Reprogramming Tumor-Associated Macrophages toward an M1 Phenotype"

_biomedicines, 2023, doi:10.3390/biomedicines11112874_

Round 1

Reviewer 1 Report

Congratulations on the present paper! 
New researches are very welcomed into a field in which not too many advances were done recently.

There are some aspects to be clarified.

1. Please explain if the study was conducted according to the Declaration of Helsinki.

2. The discussion section shoulde be extended and should incloude more comparisons with recent studies.

3. Strength and Limitations should be added.

4. The conclusion may be individualized in a separate chapter.

Author Response

Reviewer 1

Congratulations on the present paper! 
New researches are very welcomed into a field in which not too many advances were done recently.

There are some aspects to be clarified.

  1. Please explain if the study was conducted according to the Declaration of Helsinki.

Response: We thank you and the reviewers for the constructive comments. The trial was conducted according to the ethical guidelines of the Declaration of Helsinki. The use of health donor or patient-derived material has approval from the institutional research ethics committee of TMUGH (Approval Code: ZYY-IRB-SOP-016(F)-002-04; Approval Date: 2021.02.19). In addition, all patients provided signed informed consent. All animal studies were done in compliance with the regulations and guidelines of Tianjin Medical University institutional animal care (Approval Code: ZYY-DWFL-IRB-002(F)-01; Approval Date: 2021.02.19) and conducted according to the IACUC and AAALAC guidelines.

  1. The discussion section should be extended and should includemore comparisons with recent studies.

Response: We thank you and the reviewers for the constructive comments.Following the suggestion, we added related content.

  1. Strength and Limitations should be added.

Response: We thank you and the reviewers for the constructive comments.Following the suggestion, we added related content.

  1. The conclusion may be individualized in a separate chapter.

Response: We thank you and the reviewers for the constructive comments.Following the suggestion, we made the conclusion as separate chapter.

Reviewer 2 Report

In a study by Yang et al "MiR-506 promotes antitumor immune response in pancreatic cancer by reprogramming tumor-associated macrophages toward an M1 phenotype", the Authors investigated a relationship between the miR-506 and TME, especially M2-like macrophages, thus providing novel insights into mechanisms of tumor progression and potential immunotherapeutic targets for further clinical treatment. The study is technically sound, with a number of in vitro and in vivo assays, including clinical samples. 

Specific comments:

1. Please provide consistent editing e.g., in figure legends.

2. Fig. 4B, 5A,B,D,E, 6A lack statistical analysis.

3. Fig. 4 and 5A are generally of poor quality - please enlarge panels.

4. For the majority of bar graphs, please provide more detailed scale on y-axis. For example, for gene expression, there are only 0, 0.5, 1 points etc. It is difficult to estimate the values.

5. Discussion should be improved. The Authors summarize the results, and poorly put them in a broader context.

Author Response

Reviewer 2

In a study by Yang et al "MiR-506 promotes antitumor immune response in pancreatic cancer by reprogramming tumor-associated macrophages toward an M1 phenotype", the Authors investigated a relationship between the miR-506 and TME, especially M2-like macrophages, thus providing novel insights into mechanisms of tumor progression and potential immunotherapeutic targets for further clinical treatment. The study is technically sound, with a number of in vitro and in vivo assays, including clinical samples. 

Specific comments:

  1. Please provide consistent editing e.g., in figure legends.

Response: We thank you and the reviewers for the constructive comments.Following the suggestion, we fixed the relevant content.

  1. 4B, 5A,B,D,E, 6A lack statistical analysis.

Response: We thank you and the reviewers for the constructive comments.Following the suggestion, we fixed the relevant content.

  1. 4 and 5A are generally of poor quality - please enlarge panels.

Response: We thank you and the reviewers for the constructive comments.Following the suggestion, we fixed the relevant content.

  1. For the majority of bar graphs, please provide more detailed scale on y-axis. For example, for gene expression, there are only 0, 0.5, 1 points etc. It is difficult to estimate the values.

Response: We thank you and the reviewers for the constructive comments.Following the suggestion, we fixed the relevant content.

  1. Discussion should be improved. The Authors summarize the results, and poorly put them in a broader context.

Response: We thank you and the reviewers for the constructive comments.Following the suggestion, we added related content.

Reviewer 3 Report

This study aims to explore the relationship between the expression of miR-506 and the alteration of TME in PDAC patients.

I believe that the study has sufficient merit to be considered for publication on Biomedicines, although major revisions are required.

MAJOR COMMENTS

-       Abstract: Expand the abstract to include a more detailed overview of key findings, methods used, and clinical implications.

-       Introduction: To enhance the understanding of the significance of microRNAs in the diagnosis and therapy of various tumors, I suggest considering the inclusion of a brief explanation on this topic in the introduction. It could be valuable to introduce the role of microRNAs as key regulators of gene expression and mention how they have been studied in various types of tumors, including pancreatic ones. I recommend to the authors these references that I think is important and that can be of great help when modifying the manuscript (doi: 10.3390/ijms241310846, PMID 37446024).

-       Discussion. Provide a more detailed explanation of the mechanism through which miR-506 interacts with the STAT3 pathway, including key molecular steps and potential impacts on other signaling pathways.

Implications and Future Directions: Highlight clearly and tangibly how the findings could be applied in clinical practice. Describe potential scenarios where new miR-506-based therapies could enhance the efficacy of existing treatments.

-       Conclusion: Briefly highlight the main strengths of the study and the significance of your findings in relation to the broader context of pancreatic cancer research.

Moderate editing of English language required

Author Response

Reviewer 3

This study aims to explore the relationship between the expression of miR-506 and the alteration of TME in PDAC patients.

I believe that the study has sufficient merit to be considered for publication on Biomedicines, although major revisions are required.

MAJOR COMMENTS

-       Abstract: Expand the abstract to include a more detailed overview of key findings, methods used, and clinical implications.

-       Introduction: To enhance the understanding of the significance of microRNAs in the diagnosis and therapy of various tumors, I suggest considering the inclusion of a brief explanation on this topic in the introduction. It could be valuable to introduce the role of microRNAs as key regulators of gene expression and mention how they have been studied in various types of tumors, including pancreatic ones. I recommend to the authors these references that I think is important and that can be of great help when modifying the manuscript (doi: 10.3390/ijms241310846, PMID 37446024).

-       Discussion. Provide a more detailed explanation of the mechanism through which miR-506 interacts with the STAT3 pathway, including key molecular steps and potential impacts on other signaling pathways.

Implications and Future Directions: Highlight clearly and tangibly how the findings could be applied in clinical practice. Describe potential scenarios where new miR-506-based therapies could enhance the efficacy of existing treatments.

-       Conclusion: Briefly highlight the main strengths of the study and the significance of your findings in relation to the broader context of pancreatic cancer research.

Response: We thank you and the reviewers for the constructive comments.Following the suggestion, we added related content.

Round 2

Reviewer 3 Report

I believe that the study has sufficient merit to be considered for publication